# Multiverse Analysis of the Effect of a Non-Deceptive Placebo on a Neural Measure of Emotional Distress

*Hannah E. Fowles[1]\* and Peter J. Allen[1]*

[1] School of Psychological Science, University of Bristol, United Kingdom

\* dz22443@alumni.bristol.ac.uk

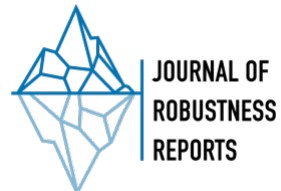

## Abstract

**An objective, neurological effect of a non-deceptive placebo observed in [1] is robust to age sub-sampling and electrode selection. However, it seems likely to be a false positive.**

## Target article

D. A. Guevarra, J.S. Moser, T.D. Wager and E. Kross, *Placebos without deception reduce self-report and neural measures of emotional distress,* Nature Communications **11**, 3785 (2020), doi:10.1038/s41467-020-17654-y

## 1    Goal

Multiverse analysis [2, 3] was used to assess how an effect observed by [1] – that a non-deceptive placebo (NDP) reduced sustained late positive potential (LPP) amplitude, which is a neural marker of emotional distress, by $2.01\mu V$ (*Hedges' g* = 0.42), $F(1, 194) = 8.98$, $p = .003$ – was influenced by alternative analytic decisions. In [1], this main effect was nested in a seven-way mixed ANOVA. It conflicts with a recent meta-analysis [4] that found no objective NDP effects in non-clinical populations ($k = 8$, $N = 583$, $g = 0.02$).

## 2    Methods

We varied the following to produce 12 unique outcomes:

- **Sample**: [1] pre-registered $N = 100$, collected data from $N = 218$ in two distinct samples, then analyzed $N = 198$. We also analyzed each sample ($n = 102$ and 96 following [1]'s exclusions) separately.

- **Age**: [1] sampled participants aged 18-31. We also sub-sampled participants between 18 and 21 years as [5-7] found differences in emotion processing and regulation between adolescence and adulthood.
- **Hemisphere**: We excluded left hemisphere electrodes as [8, 9] observed right hemisphere dominance during passive viewing tasks of emotional stimuli in a task like [1]'s and it is common practice to analyze electrode sites at which the component of interest is largest [10].

ERP (event related potential) studies often involve several multi-factor ANOVAs with many effects that are not corrected for Type I error inflation [11]. In [1], the seven-way ANOVA gave rise to 127 possible effects, all evaluated for significance at $\alpha$ = .05. As [1]'s effect is 'novel', we also used $\alpha$ = .005, as recommended by [12]. Finally, based on [1]'s plan to focus on the effects of condition and its interactions, we used a Šidák-Bonferroni adjusted $\alpha$ = .002.

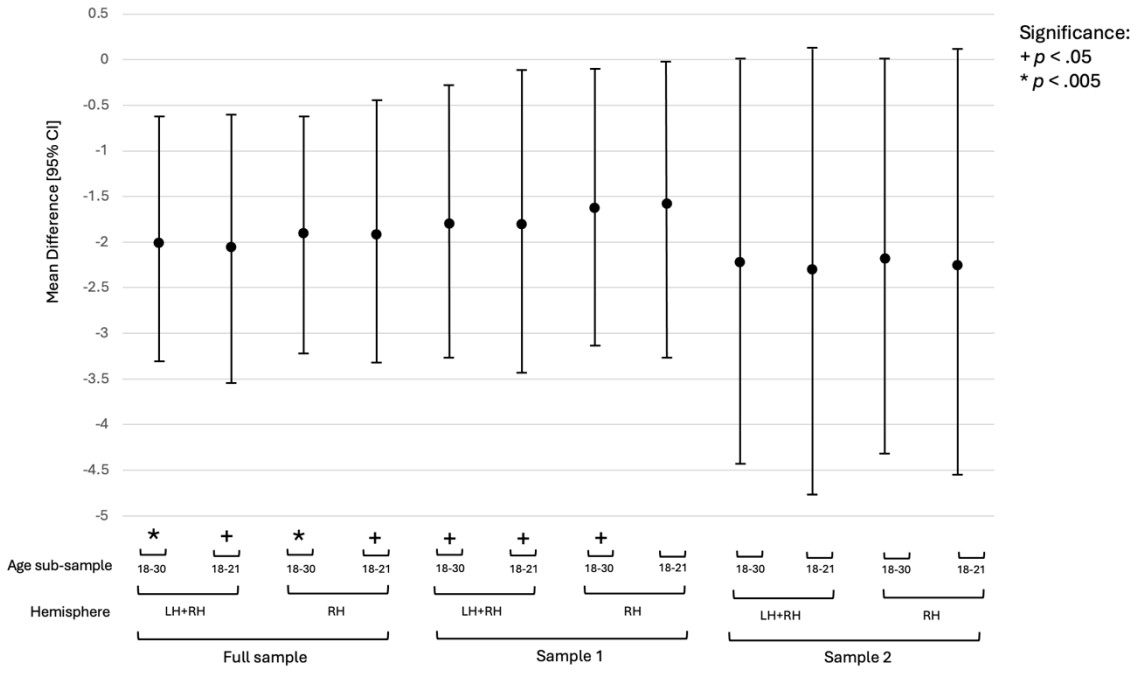

*Figure 1: Unstandardized effect sizes (mean differences) for the main effect of condition (NDP vs. control) on sustained LPP amplitude in 12 factorial ANOVAs in the multiverse analysis, evaluated for statistical significance at p < .05, .005, and .002. Mean differences < 1µV = low, > 1.5µV = robust, and > 3µV = very robust, according to [13]. Sample sizes ranged from N = 83 to 198. LH = Left Hemisphere; RH = Right Hemisphere.*

## 3    Results

Of the 12 effects in Figure 1, seven were statistically significant at $p < .05$, two were significant at $p < .005$ and none were significant at $p < .002$. The direction and magnitude of the effects, which ranged from -1.58µV to -2.30µV ($g$ = 0.40 to 0.47), remained consistent across the multiverse. The effect was robust against the age sub-sample and hemisphere selection alternatives. However, wide CIs with lower bounds close to zero were routinely observed, indicating uncertainty regarding the true effect size.

# 4    Conclusion

The NDP effect observed in [1] appears robust to age sub-sampling and electrode selection. However, had [1] stuck to their original sampling plan and taken a reasonable approach to Type I error control, it is likely that their overall conclusion would have been more consistent with [4]. That is, that NDPs do not demonstrate reliable objective effects in non-clinical populations. In this context, it seems possible that the effect observed by [1] is a false positive. However, this requires further investigation with a high-quality replication.

## Acknowledgments and Disclosures

**Reproducibility** The original effect was successfully reproduced using SPSS syntax accompanying [1]. It is located at https://osf.io/s3b8d/ (see Experiment 2 Data Files).

**Code and Data Availability** SPSS syntax for reproducing this multiverse analysis is available at https://osf.io/vxuhc/. The original data are available at https://osf.io/s3b8d/ (see Experiment 2 Data Files).

**Pre-Registration** This multiverse was part of a larger study (https://doi.org/10.17605/osf.io/vxuhc) pre-registered at https://osf.io/vxuhc/files/6d3j9. Note that we did not pre-register separate analyses for Sample 2.

**Author Contributions H.E.F.**: Conceptualisation, Data curation, Formal analysis, Investigation, Methodology, Visualisation, Writing - original draft, and Writing – review & editing. **P.J.A.**: Conceptualisation, Supervision, and Writing - review & editing.

**Funding** The authors received no financial support for the research, authorship, or publication of this article.

**Conflicts of Interest** The authors have no competing or conflicting interests to declare.

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
