# Peer review of "Multiverse Analysis of the Effect of a Non-Deceptive Placebo on a Neural Measure of Emotional Distress"

_Journal of Robustness Reports_

## Round 1 · Referee Report · Anonymous (Referee 1) · 2025-8-19

Report

Fowles and Allen report a slice of a multiverse re-analysis of a clinical study. While the topic is of interest, I do have concerns about the validity of the arguments put forward. Taking the results of the analysis at face value, I am marginally more confident in the robustness of the original result. I do not believe the re-analysis as presented here permits any conclusions about clinical significance.

Major issues

  • I am not convinced by the finding that the p-value is sensitive to the sample size. With a sufficiently broad range of sample sizes, any p-value can be achieved. That is a well known property of p-values and should probably not be used to support a substantive conclusion. Similarly, it is not a strong criticism to point out that p-values are sensitive to p-value correction.
  • The authors conclude that the effect is likely of low clinical significance, but offer no specific argument for this conclusion. I found no unstandardized effect sizes to evaluate. Is the effect small, in an absolute sense, compared to other effects? Is there a subject matter consensus on effect size expectations? If such benchmarks exist, the authors could make that explicit in support of this conclusion. Either way, unstandardized effect sizes seem more appropriate for this purpose.

Minor issues

  • The report uses the occasional bit of undefined jargon (e.g., the 'exploratory p-value' in the figure caption is not explained anywhere).
  • Between the three vertical axes, inverted scales, and dense horizontal axis, the figure is difficult to read. Also, the connecting lines between markers give an incorrect suggestion of continuity. Perhaps this is a job for the more conventional box-and-whisker plots with stars (if you decide that p-values are still interesting to report here).

Very minor issue

  • I understand this is probably the journal's style and not a decision by the authors, but the citation style uses the same format for in-text and parenthetical references, which gets a little confusing.

Suggestions

  • I would encourage the authors to call balls and strikes, especially regarding the effects of downsampling the data or correcting the p-value for multiple testing. If those strategies are the right ones to take, then other strategies do not need to clutter the report. If there are subjective aspects to the decision, then they could note what those are.
  • The Age and Hemisphere dimensions seem the most relevant for a robustness check.
  • The authors mention a Bayesian angle to the full multiverse analysis, which might be more compelling than focusing on p-values.

Recommendation

Ask for major revision

  • validity: -
  • significance: -
  • originality: -
  • clarity: -
  • formatting: -
  • grammar: -

Author:  Peter Allen  on 2025-11-04  [id 5990]

(in reply to Report 1 on 2025-08-19)
Category:
answer to question

Thanks for taking the time to review this manuscript. Our responses to your comments are as follows, and you'll be able to see the changes that we have made in response to them in our revised manuscript, which I will upload after also responding to R2:

Major issues:

1. In the revised manuscript we have shifted the interpretation to effect sizes (both unstandardized and standardized) and whether they cross three reasonable/justifiable thresholds for statistical significance. We feel strongly that some Type I error control should have been employed in the Guevarra et al. (2020) study, and two modestly conservative approaches both suggest a conclusion opposite to the one made by the original authors.
2. Fair call. We’ve removed all references to clinical/practical significance and reported both unstandardized and standardized effect sizes. The original effect is actually large relative to all the other effects included in the Spille et al. (2023; https://doi.org/10.1038/s41598-023-30362-z) meta-analysis (the next largest is around half the size, and the others are either negative or close to zero). This, combined with the point above about Type I error control lead to the possibility that the effect reported by Guevarra may be a false positive. This is the conclusion that we’ve emphasised in the revised manuscript.

Minor issues:

1. This language is no longer necessary and has been dropped from the revised manuscript.
2. We agree that the graph was overly complex. We’ve stripped it back to unstandardized effect sizes and their confidence intervals, along with flags to indicate whether they cross the significance thresholds noted above.

Very minor issue:

1. This is the journal’s style.

Suggestions:

1. We feel that this point is addressed in the revised manuscript. There are now four main points which are hopefully clearer: (a) the effect sizes are robust to the age and hemisphere alternatives; (b) sticking to the original sampling plan combined with modest Type I error control changes the conclusion one draws about the reliability of the effect; (c) the effect is an outlier relative to others in this space; and (d) due to (b) and (c), it seems likely to be a false positive. Of course, a high-quality replication is needed before a firm conclusion can be drawn about (d), which is the point we now finish on.
2. These are now more prominent in the results, although we do feel that the Type I error control issue is important in terms of being able to interpret these.
3. We agree. However, the 7-way Bayesian ANOVAs proved impossible to run. We attempted R and JASP on several computers but had no luck. In the report on which this manuscript is based, we reduced them down to 3-way ANOVAs. However, this leaves us unable to directly compare them with the 7-way frequentist models reported in the manuscript. Consequently, we have elected not to report them here.

---

## Round 1 · Referee Report · Anonymous (Referee 2) · 2025-9-3

Report

The article presents a multiverse analysis to assess the robustness of Guevarra et al. (2020)'s finding that non-deceptive placebos reduce neural measures of emotional distress. Three analytical decisions —sample size (N=198 vs N=100), age range (18-31 vs 18-21 years), and electrode selection (all vs right hemisphere)— are systematically varied at three alpha levels. The approach addresses an important methodological question about analytical flexibility in ERP research.

Major: - The paper assumes complete familiarity with multiverse analysis without adequate introduction. I understand that word limits preclude expansive explanations, but I think for a broader readership, multiverse analysis requires some explanation of its logic and interpretation principles. Readers unfamiliar with the approach cannot follow why testing multiple specifications matters or how to interpret the resulting distribution of p-values. The Steegen et al. reference of course provides some context but you would want the paper to be possible to be read stand-alone. - The target effect is never explicitly stated, making it difficult to evaluate replication success. The authors reference "the main effect of condition on sustained LPP amplitude" but never specify exactly what Guevarra et al. found. Without this baseline, I had a hard time assessing whether 7/16 significant results at p < .05 represents successful or failed replication. - The sample size reduction seems to contradict the stated goal of detecting small effects with stringent alpha levels. The authors cite Gibney et al. (2020) to justify N=100, but that paper's simulations show sufficient power to detect 0.5μV effects with 100 participants. The original Guevarra effect appeared to be approximately 0.4μV. Reducing sample size below what is required for the detection of the previously observed effect, seems to make it very unlikely that you would find significant effects at even stricter alpha levels. - The framing consistently emphasizes negative aspects while downplaying successful replication. The abstract concludes "limited practical significance" despite replicating the original effect in nearly half the specifications, including the exact original analysis. I agree that multiple testing is a thing and is important, but the multiverse results could also be taken to demonstrate that the effect exists robustly across reasonable analytical choices. It gives the impression that the authors were unhappy with the original effect, and were perhaps convinced that it would not be robust. - Figure 1 is a very busy graph. Perhaps it is a standard graph in some fields, but since the authors are trying to make a simple point, with a simple short article, I think a simpler graph might be possible.

Minor: - The authors subsample 18-21 year-olds citing "differences in emotion processing and regulation between adolescence and adulthood" but provide no specific rationale for why placebo effects would differ across this age range, nor why this represents a reasonable analytical choice rather than an arbitrary restriction. - The authors choose to do an analysis excluding left hemisphere electrodes citing "right hemisphere dominance during passive viewing tasks" but don't explain whether this represents a reasonable analytical choice or a theoretical claim that should be tested separately. - The conclusion that "clinical utility appears limited" extends beyond what the analysis can support. Multiverse analysis can demonstrate analytical robustness but cannot assess clinical significance, which depends on a lot of other factors. - The title could provide more information about the specific finding. - "partially pre-registered" raises more questions than it needs to.

Recommendation

Ask for major revision

  • validity: high
  • significance: ok
  • originality: ok
  • clarity: good
  • formatting: good
  • grammar: excellent

Author:  Peter Allen  on 2025-11-04  [id 5991]

(in reply to Report 2 on 2025-09-03)
Category:
answer to question

Thanks for your comments. We have addressed them in the order they were made:

Major -

Multiverse: We are very constrained by the word-limit and think that readers of this journal are likely to be familiar with this technique. In the revised manuscript we’ve added in a reference to Steegen et al. (2016) as well as a very recent paper by Heyman et al. (2025). We feel that these are good starting points for anyone who is unfamiliar with the idea of a multiverse analysis.

Target effect: This is now explicit in the introduction of the revised manuscript.

Sample size: The important point here is that the original authors pre-registered N = 100 but then collected data from over twice this number of participants, in two distinct phases. We do like the idea of saying “let’s imagine that their sampling decisions were based on a power analysis”, but the approach we took to this was flawed. To do this properly we should have first imagined several reasonable power analyses (rather than just one). We could have then dropped cases randomly over repeated iterations of the analysis and averaged the results of these. However, this would have been a distraction from the more important point, which is if the original authors had stuck to their original sampling plan and performed a reasonable level of Type I error control, their conclusions would likely have been consistent with the recent meta-analysis performed by Spille et al. (2023), who found no reliable objective effects of non-deceptive placebos in non-clinical populations.

Framing: In the revised manuscript we have dropped references to practical significance throughout and re-framed the conclusions to emphasise how (a) the effect sizes are robust to the age and hemisphere alternatives; (b) sticking to the original sampling plan combined with modest Type I error control changes the conclusion one draws about the reliability of the effect; (c) the effect is an outlier relative to others in this space; and (d) due to (b) and (c), it seems likely to be a false positive. Of course, a high-quality replication is needed before a firm conclusion can be drawn about (d), which is the point we now finish on.

Figure 1: We’ve substantially simplified this in the revised manuscript. Specifically, we’ve stripped it back to unstandardized effect sizes and their confidence intervals, along with flags to indicate whether they cross three significance thresholds (p < .05 per the original authors; p < .005 as recommended for ‘novel’ findings by Benjamin et al., 2018; and a Šidák-Bonferroni adjusted p < .002).

Minor –

Age sub-sample: The point of this analytic alternative is that there are differences in emotion processing between adolescence and adulthood, which in turn may have impacted the effect of the NDP. Running the analysis on a sub-sample of 18–21 year-olds could provide insight into whether the effect of condition (NDP) on sustained LPP amplitude is sensitive to differences in emotion processing and regulation between younger and older participants (As it happens though, it was not.)

Hemisphere: Excluding left hemisphere electrodes from the analysis represents a reasonable analytic choice, as it is common practice in ERP studies to analyse electrode sites at which the component of interest is largest. Since previous research has observed right hemisphere dominance in emotion processing (Gainotti, 2019; Hartikainen, 2021) and in passive viewing tasks of emotional stimuli (Kayser et al., 2000; Zhang & Zhou, 2014) it seems reasonable that Guevarra et al., (2020) could have chosen to focus on the right hemisphere only. (This also would have removed one of the 7 factors from the ANOVA, somewhat reducing the chances of a type I error.)

Clinical utility: In the revised manuscript we’ve removed references to this, as described above under ‘framing’.

Title: This is difficult given the strict character limits. As the key findings will appear directly below in the two-sentence abstract, and there seems little point in duplicating this in the title, we’ve decided to leave it as it is.

Pre-registered: We’ve removed this and placed it in the disclosures section.

---

## Round 2 · Author Response

Thank you to both reviewers for their comments on the first iteration of this manuscript. We have substantially revised the manuscript in response to these. A summary list of changes is provided below. Regards, Peter Allen and Hannah Fowles

---

## Round 2 · List of Changes

- We have made the target effect more explicit in the introduction, by including all relevant statistics.
- We have included two references for multiverse, including a very recent paper by Heyman et al. (2025).
- We have removed the N = 100 analytic alternatives. A full rationale for this is provided in our response to R2.
- We have dropped references to practical/clinical significance throughout and re-framed the conclusions to emphasise how (a) the effect sizes are robust to the age and hemisphere alternatives; (b) sticking to the original sampling plan combined with modest Type I error control changes the conclusion one draws about the reliability of the effect; (c) the effect is an outlier relative to others in this space; and (d) due to (b) and (c), it seems likely to be a false positive. Of course, a high-quality replication is needed before a firm conclusion can be drawn about (d), which is the point we now finish on.
- We’ve substantially simplified the graph. Specifically, we’ve stripped it back to unstandardized effect sizes and their confidence intervals, along with flags to indicate whether they cross three reasonable/justifiable significance thresholds.
- We have also made a number of more modest changes, as detailed in our responses to each reviewer.

---

## Editorial Decision

accepted_in_target_journal